# Design of γ-Alumina-Supported Phosphotungstic Acid-Palladium Bifunctional Catalyst for Catalytic Liquid-Phase Citral Hydrogenation

**Abdul Karim Shah [1,*], Syed Nizam-uddin Shah Bukhari [2], Ayaz Ali Shah [3,*], Abdul Sattar Jatoi [1], Muhammad Azam Usto [1], Zubair Hashmi [1], Ghulam Taswar Shah [1], Yeung Ho Park [4], Moo-Seok Choi [4], Arshad Iqbal [1], Tahir Hussain Seehar [3] and Aamir Raza [3]**

[1] Department of Chemical Engineering, Dawood University of Engineering and Technology, MA Jinnah Road, Karachi 74800, Pakistan

[2] Department of Basic Science and Humanities, Dawood University of Engineering and Technology, MA Jinnah Road, Karachi 74800, Pakistan

[3] Department of Energy and Environment Engineering, Dawood University of Engineering and Technology, MA Jinnah Road, Karachi 74800, Pakistan

[4] Fine Chemical Process Laboratory, Department of Chemical Engineering, Hanyang University, Erica Campus, Sangnok-gu, Ansan 15588, Korea

\* Correspondence: abdulkarim.shah@duet.edu.pk (A.K.S.); aas@duet.edu.pk (A.A.S.)

**Abstract:** This study primarily addresses the development of dynamic, selective and economical metal–acid (bifunctional) catalysts for one-pot menthol production by citral hydrogenation. Specifically, various metals such as Pd, Pt, Ni, Cs and Sn were doped over alumina support. Additionally, bifunctional composite catalysts were also prepared with the impregnation of heteropoly acids and Pd precursors over alumina support. Analytical techniques (e.g., BET, PXRD, FT-IR, pyridine adsorption and amine titration methods) were applied for characterization of the most efficient and selective catalysts (e.g., $Al_2O_3$ and PTA-Cat-I). Similarly, most of the essential operational variables (e.g., loading rate of metal precursor, type of heteropoly acid, temperature, gas pressure and reaction time) were examined during this study. The experimental data shows that the bifunctional catalyst (PTA-Cat-I) produced 45% menthol at full citral substrate conversion (r = 0.038 mmoles.min$^{-1}$) in liquid-phase citral hydrogenation (at optimized operating conditions: 70 °C, 0.5 MPa and 8 h). However, the heteropoly acid-supported bifunctional catalysts (e.g., PTA-Cat-I, PMA-Cat-I, SMA-Cat-I and STA-Cat-I) resulted in cracking and the dehydration of isopulegol/menthol by the generation of side products (e.g., 4-isopropyl-1-methyl, cyclohex-1-ane/ene); therefore, menthol yield was extensively diminished. On the other hand, non-acidic catalysts (e.g., Cat-I, Cat-II, Cat-III, Cat-IV and Cat-V) readily promoted hydrogenation reactions. The optimum menthol yield occurred due to the presence of strong Lewis and weak Bronsted acid sites. Mass transfer and reaction rate were substantially diminished due to acidity strength, heteropoly acid type and blockage of pores by the applied bifunctional catalysts.

**Keywords:** alumina; heteropoly acids; bifunctional catalysts; citral hydrogenation; menthol

## 1. Introduction

Citral (3,7 dimethyl-2,6 octadienal) is an unsaturated aldehyde compound that is normally present in lemongrass oil. It possess three hydrogenation sites: a carbonyl group, a double bond conjugated with the CO group and an additional isolated C-C bond [1,2]. This type of unsaturated aldehyde has been studied widely for hydrogenation processes involved in fine chemical production [3]. The hydrogenation of the citral molecule provides multiple products such as unsaturated alcohols (nerol/geraniol), cyclic compounds (isopulegol and menthol) and hydrogenated products (citronellal and dihydrocitronellal) [4,5].

From previous literature and current market demand, it is clear that menthols have high market demand due to their variety of applications in pharmacy, cosmetics and the tobacco industry [6–8]. Menthol is produced via natural and synthetic routes. The contribution of natural and synthetic sources is approximately 80% and 20%, respectively [9–11]. Among menthol synthesis routes [12], citral hydrogenation is an easy and comfortable route. It consists of three reaction steps: (a) hydrogenation of citral to citronellal, (b) cyclisation of citronellal to isopulegol and (c) hydrogenation of isopulegol to menthols, as shown in Scheme 1. Mäki-Arvela et al. [10], prepared metal–acid catalysts by doping metals (Ni and Pd) on H-MCM-41 material and tested them in liquid-phase hydrogenation of citral, producing 54% menthol at full conversion of citral substrate. Menthol synthesis from citral hydrogenation is a more complex and multistage reaction. A number of side products have been generated due to improper design of bifunctional catalysts, which results in less menthol formation [13]. In the last decade, a number of bifunctional catalysts have been designed for citral hydrogenation and their catalytic performance in menthol synthesis has been evaluated [7,14,15]. Salminen et al. [9], obtained about 5% menthol selectivity in liquid-phase hydrogenation of citral using a bifunctional catalyst. Correspondingly, Virtanen et al. [16] tested Pd/Zn-supported ionic liquid material in liquid-phase citral hydrogenation and obtained 30% menthol yield at full citral substrate conversion.

Additionally, numerous catalysts have been developed for the citral hydrogenation to menthol reaction [16,17], with most catalysts having been shown to be highly active in the hydrogenation reaction, but less selective for menthol production. However, most catalysts have been found to be non-recyclable. Low menthol synthesis and catalyst regeneration may be the main problems that are being addressed by authors of bifunctional catalyst designs. Therefore, more experimental study is necessitated for the design of active, selective and recyclable bifunctional catalysts for one-pot menthol synthesis.

Previous research has acknowledged that most acidic catalysts possess various ratios of Lewis and Bronsted acid sites on their surfaces [17–22]. Moreover, various kinds of inorganic (e.g., HCl, $H_2SO_4$, $HNO_3$) and organic acids (e.g., $CH_3COOH$, $C_3H_6O_3$, $CH_2O_2$) have been applied in the design of acid catalysts materials [23,24]. The abovementioned acids have limited boundaries (e.g., lower activity, side reactions, leaching, toxicity and regeneration) in the field of catalysis. Simultaneously, heteropoly acids (e.g., HPAs) have revealed greater prospects in the area of catalysis. Usually, these heteropoly acids have been thought of as strong acids, as they have a higher amount of acidity and catalytic activity [13,23,24]. In the heteropoly acid (HPAs) family, phosphotungstic acid (PTA) is the most stable and acidic, compared to silicon-tungstic acid (STA), phosphor-molybdic acid (PMA), silicon-molybdic acid (SMA) and so on [25,26]. It has shown that heteropoly acid applications are limited because of their lower surface area characteristics [26]. Previous literature has recommended that these HPA limitations (e.g., lower surface area prohibits mass transfer and diffusion of reactant molecules in the catalyst's pores) can be resolved through the dispersion of HPAs over non-acidic and micro-mesoporous supports. This mechanism could possibly help in enhancing mass transfer and reaction rate. Considering the surface area characteristics of heteropoly acids, alumina material can be used as a non-acidic and mesoporous support [27,28]. It is probably beneficial to maintain the catalytic, thermal and acidic characteristics of HPAs by diffusing them over an exterior surface of alumina material [29]. This mechanism would provide high surface area, mass transfer and thermal stability [30–34]. Alumina-supported heteropoly acid catalysts have been applied in the esterification of alcohols [35]. Rb- and Cs-doped STA-supported alumina catalysts have been applied in dehydration of glycerol to acrolein [36]. HPA-supported alumina catalyst was applied in trans/esterification of high acid feed stocks [37]. To date, metal-doped HPA-supported alumina acid as well as bifunctional catalysts have not been investigated in liquid-phase citral hydrogenation and the citronellal cyclisation reaction for obtaining menthol and isopulegol products. Our current research has primarily concentrated on the preparation of bifunctional catalysts and determining their catalytic performance under optimized process parameters.

**Scheme 1.** Reaction scheme of multistage liquid-phase citral hydrogenation and possible side reactions (e.g., dehydration and cracking reactions).

In this research, the main objective was to design a bifunctional (metal–acid) catalyst by using dispersing metal and heteropoly acids over alumina. The catalyst was applied in liquid-phase citral hydrogenation to obtain higher menthol production. In the initial stage, various metal types were doped over alumina and tested in citral hydrogenation. Furthermore, metal–acid-supported alumina catalysts were designed through the ion-exchange method and tested in citral hydrogenation under optimized process conditions. In this work, we have optimized bifunctional catalyst design, reaction operating parameters, catalytic activity and selectivity. Additionally, the reaction kinetics of the liquid-phase citral hydrogenation and citronellal cyclisation reactions were measured. This study may possibly be helpful for selection of the most applicable and cost-effective bifunctional catalysts for citral hydrogenation for menthol synthesis.

## 2. Material and Methods

### 2.1. Materials

Palladium(II) chloride (99.9% pure), palladium(II) acetylacetonate (99%), nickel nitrate hexahydrate (99%), platinum(II) acetylacetonate (99% pure) cesium chloride (99%), tin(II) chloride (99%), cesium chloride (99%), 12-phosphotungstic acid, silicotungstic acid, silicomolybdic acid hydrate, phosphomolybdic acid, ($\pm$) citral (95% pure), alumina oxide (neutral), silica oxide (fumed), zeolite beta hydrogen, ethyl alcohol (99.9% pure), methyl alcohol (99.9% pure), dichloromethane (95% pure), toluene, cyclohexane, n-hexane and 1,4-dioxane were purchased from Sigma Aldrich. Cyclohexane and n-hexane solvents were purchased from Dae-Jung Chemicals. Other supporting chemicals such as methanol and nitrobenzene were purchased from Junsei Chemicals.

### 2.2. Preparation of Bifunctional (Metal–Acid) Catalysts

According to the previous literature review, it has been observed that various metals, as well as heteropoly acid-supported bifunctional catalysts have been prepared for innumerable applications [24,38–43]. Therefore, in this work, the alumina and heteropoly acid-supported bifunctional metal catalysts have been prepared using dry impregnation and ion-exchange techniques [44]. In the first stage, alumina-supported metal catalysts such as 5 wt.% Pd-$Al_2O_3$ (Cat-I), 5 wt.% Pt-$Al_2O_3$ (Cat-II), 5 wt.% Ni-$Al_2O_3$ (Cat-III), 5 wt.% Cs-$Al_2O_3$ (Cat-IV) and 5 wt.% Sn-$Al_2O_3$ (Cat-V) were synthesized by using the wet impregnation process in accordance with published literature. Five grams of dried and washed $\gamma$-alumina support was poured into 20 mL ethanol solvent and 5 wt.% of all aqueous metal precursors (Pd, Pt, Ni, Sn and Cs) was doped over alumina support using the wet impregnation method at room temperature for 24 h at a constant stirring rate. After impregnation of metals over alumina, all samples were dried in a vacuum oven at 60 °C for 12 h. Furthermore, the prepared metal-supported samples were activated by using $H_2$ gas (30 mL.min$^{-1}$) at high temperatures (250–550 °C). In the second stage, heteropoly acid (HPA)-supported alumina-based bifunctional metal acid catalysts were prepared by using the wet impregnation method. The 20 wt.% of phospho-tungstic acid (PTA-Cat-I), silicon-tungstic acid (STA-Cat-I), silicon-molybdic acid hydrate (SMA-Cat-I) and phospho-molybdic acids (PMA-Cat-I) were impregnated on alumina-supported catalyst (Cat-I) using toluene solvent [12] and were reduced at 250 °C for 2 h using $H_2$ gas (30 mL.min$^{-1}$) in a glass tubular furnace. Finally, the reduced catalysts (e.g., Cat-I, Cat-II, Cat-III, Cat-IV, Cat-V, PTA-Cat-I, PMA-Cat-I, STA-Cat-I and SMA-Cat-I) were immediately used in the liquid-phase citral hydrogenation reaction.

### 2.3. Catalysts Characterization

XRD patterns of HPA-supported alumina-based bifunctional metal acid catalysts analyzed with the help of an X-ray diffraction meter (XRD-6000, Shimadzu, Kyoto, Japan) with an accelerating voltage of 40 kV and applied current of 100 mA. However, bulk elemental composition was ascertained with the help of inductively coupled plasma-optical emission spectrometry (ICP-OES). For elemental analysis, the bifunctional catalysts (e.g., W, P and Pd)

were investigated and quantified based on our previous methodology [25,29]. The TriStar II 3020 2.00 at $-196\ ^\circ$C was applied to determine surface area (BET) and pore volume distribution (BJH) (operating conditions: degassed at 300 $^\circ$C for 2 h) of the prepared catalysts [26].

Likewise, Fourier transform infrared (FTIR) analysis (FTIR spectrometer Nicolet, iS10, conditions: 400–4000 cm$^{-1}$, optical resolution of 8 cm$^{-1}$ with 32 scans) was used to investigate the existence of functional groups and structural compositions of the materials. Furthermore, the pyridine adsorption method was used to examine Lewis and Bronsted acid sites of newly synthesized catalyst materials; the self-supported KBR wafers (1 cm$^2$, 11 tons cm$^{-2}$ and 30 mg) were prepared and totally dried into specially designed stainless steel IR cells by using a vacuum pump at 400 $^\circ$C for 2 h. The operating procedure for pyridine adsorption was followed as described in our previous articles [30,45,46].

Similarly, the amine titration method (0.1 N of weak base (n-butyl amine) and indicator (dimethyl yellow) chemical reagents) was used to evaluate the acidic properties of the newly synthesized catalysts [29].

### 2.4. Catalytic Reaction Study

2.4.1. Citronellal Cyclisation to Isopulegol

In this citronellal cyclisation reaction, citronellal (95% pure), benzene solvent, nitrobenzene (internal standard) and freshly prepared catalysts were used. This cyclisation reaction was performed into an autoclave reactor with a magnetic stirrer and sampling port/tube. The citronellal cyclisation reaction was operated at fixed parameters (e.g., temperature 70 $^\circ$C, N$_2$ pressure 0.5 MPa, 300 rpm). The detailed reaction performance procedure is mentioned in our already published work [24].

2.4.2. Synthesis of Menthols via Liquid-Phase Citral Hydrogenation

The hydrogenation reaction of citral substrate was performed in a stainless-steel autoclave reactor (50 mL capacity containing 4.5 mmol of citral substrate, 25 mL solvent, 0.2 mL nitrobenzene and 0.2 g newly activated catalyst). The hydrogenation reaction was performed at fixed operating conditions (e.g., temperature 70 $^\circ$C, H$_2$ pressure 0.5 MPa and 300 rpm).

Gas chromatography (GC-Agilent, model no. 7890A) with a chiral column (Cyclodex-B (Agilent), length 60 m, diameter 0.254 mm and film thickness 0.25 μm) was used to analyze the products and side reaction products of the collected samples. GC operating conditions were the same as described in previous literature [24,29]. Additionally, the obtained GC product peaks were compared with GC standard chemicals and the GC–MS technique.

## 3. Results and Discussion

### 3.1. Catalysts Characterization

In this study, we designed Pd-HPA-alumina (e.g., PTA-Cat-I; STA-Cat-I; SMA-Cat-I; PMA-Cat-I) and other simple metal (e.g., Pd, Pt, Ni, Sn and Cs)-supported alumina (Cat-I, Cat-II, Cat-III, Cat-IV, Cat-V) catalysts. Herein, we selected important catalysts (e.g., based on catalytic performance) for the characterization study. We will here discuss their characterizations in detail. [33]. XRD analysis of the selected catalysts is shown in Figure 1. According to XRD data, alumina support is in the amorphous phase. Some changes in the XRD pattern (e.g., new XRD peaks appeared at 33 and 54) were observed after PTA and Pd impregnation on alumina. The material (PTA-Cat-I) shows an X-ray diffraction pattern with incipient signals of very wide bases that are characteristic of materials with poor crystallinity (Figure 1). However, the BET surface area and pore volumes of alumina support were 136 m$^2$.g$^{-1}$ and 0.248 cm$^3$.g$^{-1}$, and decreased to 120 m$^2$.g$^{-1}$ and 0.164 cm$^3$.g$^{-1}$, respectively, after impregnation of phosphotungstic acid (20 wt.%) and Pd loading (5 wt.%) (Table 1). The adsorption–desorption intensity of alumina decreased after impregnation of PTA and Pd metal (Figure 2). Nonetheless, the doping of PTA and Pd particles over alumina support probably reduced some portion of the surface area of the alumina. Furthermore, the acidity characteristic of the catalysts was investigated

using the amine titration technique. ICP-OES elemental analysis confirmed the doping of phosphotungstic acid over alumina support (Cat-I) (Table 1).

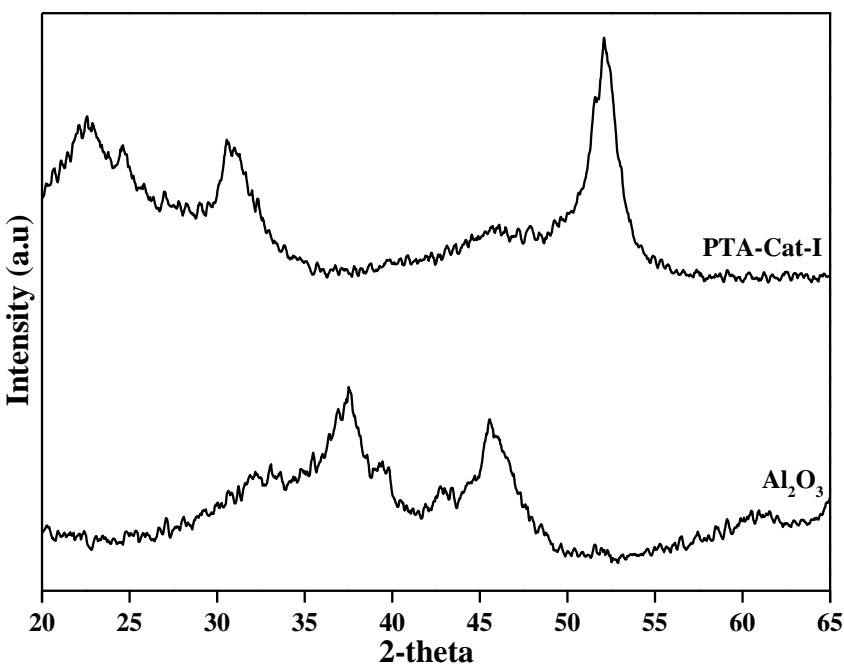

**Figure 1.** XRD patterns of $Al_2O_3$ and PTA-Cat-I catalyst samples.

**Table 1.** Catalyst characterization of support and acidic catalysts.

| Catalyst Type | BET ($m^2.g^{-1}$) [a] | VP ($cm^3.g^{-1}$) | dp (nm) | Acidity [b] |
|---|---|---|---|---|
| $Al_2O_3$ | 136 | 0.248 | 7.2 | n.d |
| * PTA-Cat-I | 120 | 0.164 | 5.4 | 0.51 |

[a] $S_{BET}$ and $Vp_{tot}$ were determined through the BET equation. [b] was determined by the amine titration method. * ICP-OES data confirm the presence of phosphorous (0.341 mg/L) and tungsten (3.115 mg/L) in Cat-I (phosphotungstic acid-supported alumina).

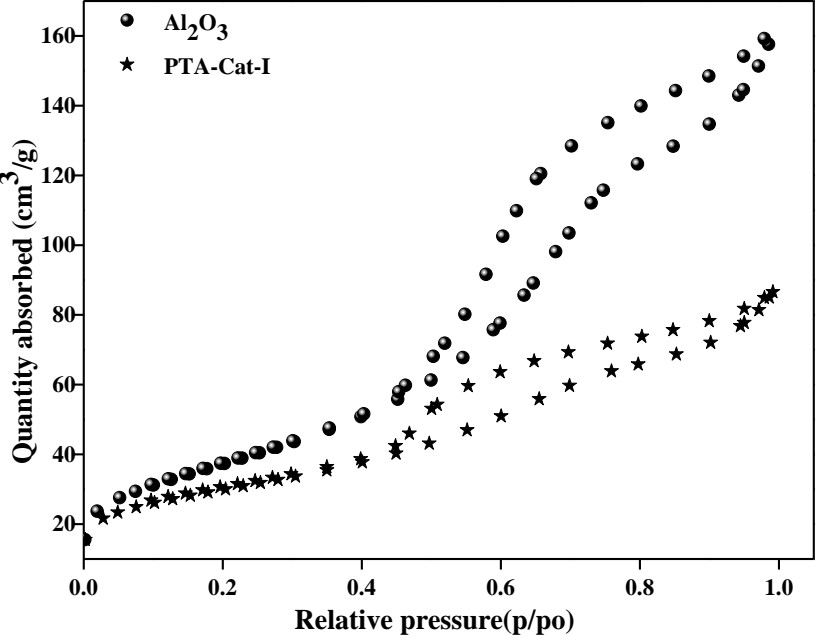

**Figure 2.** BJH pore size distribution plots and $N_2$ adsorption–desorption isotherms of $Al_2O_3$ and PTA-Cat-I catalyst samples.

The data in Table 1 suggest that alumina support is neutral (no acidic characteristic), but that its acidity was enhanced with the impregnation of phosphotungstic acid (e.g., 0.51). The acidic property of alumina was developed with a combination of heteropoly acid ions.

Figure 2 shows adsorption–desorption isotherm characterization of alumina and HPA-supported alumina catalysts. The sorption study shows that alumina is partial mesoporous, and its mesoporosity characteristics decreased with the impregnation of HPA and Pd over alumina.

Figure 3 shows the FTIR spectra of alumina and PTA-Cat-I catalyst samples. With impregnation of Pd and phosphotungstic acid over alumina, some changes (e.g., peaks at wavenumbers 1600 and 3600 cm$^{-1}$ were decreased, and new peaks appeared at wavenumbers 600 and 1000 cm$^{-1}$) were observed in the FTIR spectra. These changes might show the decrease of water and hydroxyl group peaks (e.g., probably a decrease in Bronsted acidity).

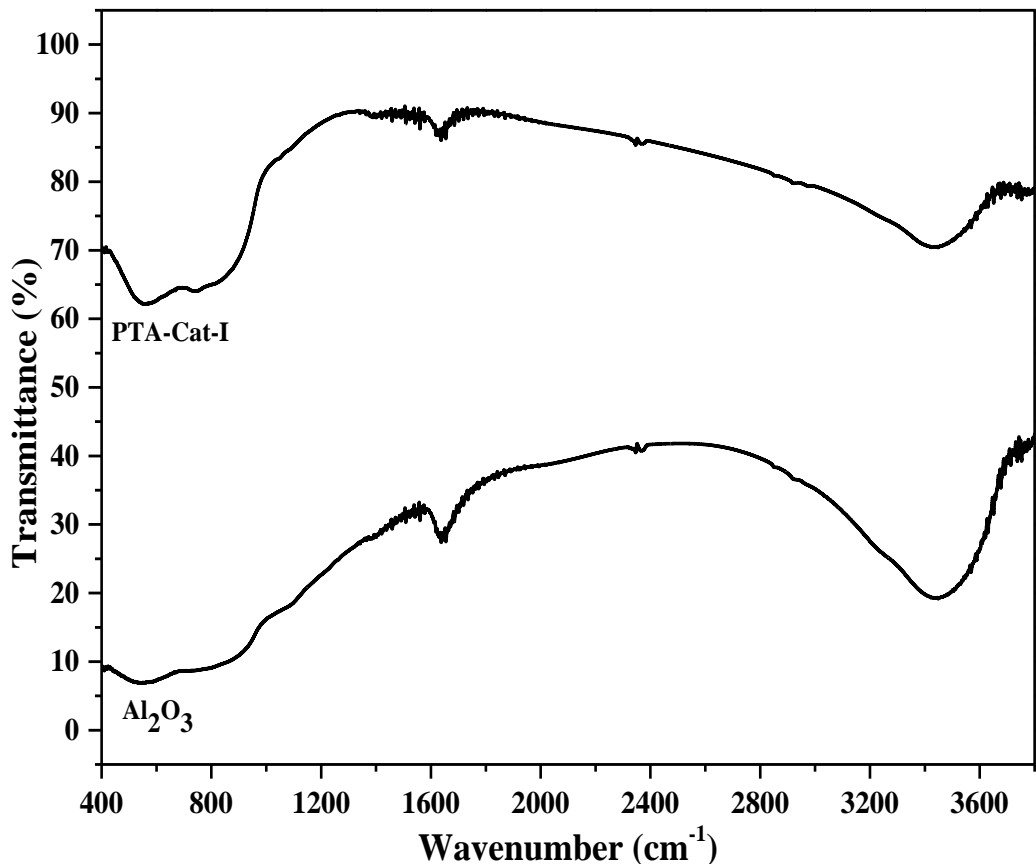

**Figure 3.** FTIR spectra of Al$_2$O$_3$ and PTA-Cat-I catalyst samples.

Additionally, the concentration of Lewis and Bronsted acid sites of samples was examined using the pyridine adsorption technique. Pyridine adsorption data imply (Figure 4) that alumina has very weak Lewis acidity; it was improved with the doping of phosphotungstic acid. Additionally, pyridine adsorbed peaks (e.g., 1440 and 1598 cm$^{-1}$) of alumina increased after impregnation with phosphotungstic acid. The increase in peak intensity shows an increase in Lewis acidity. The low intensity of Bronsted acidity was determined from small peaks of pyridine adsorption (e.g., 1490 and 1530 cm$^{-1}$). The Pd-PTA- alumina (e.g., PTA-Cat-I) possess a higher amount of Lewis acid sites over the catalyst surface in the presence of weak Bronsted acid sites. Furthermore, it was noted that bifunctional catalyst (PTA-Cat-I) normally possesses strong Lewis and weak Bronsted acid sites on the surface of the catalyst, as presented in Figure 4. In addition, the concentration of Lewis (L) and Bronsted acid sites (B) of prepared catalyst samples was calculated by pyridine adsorbed peaks [45].

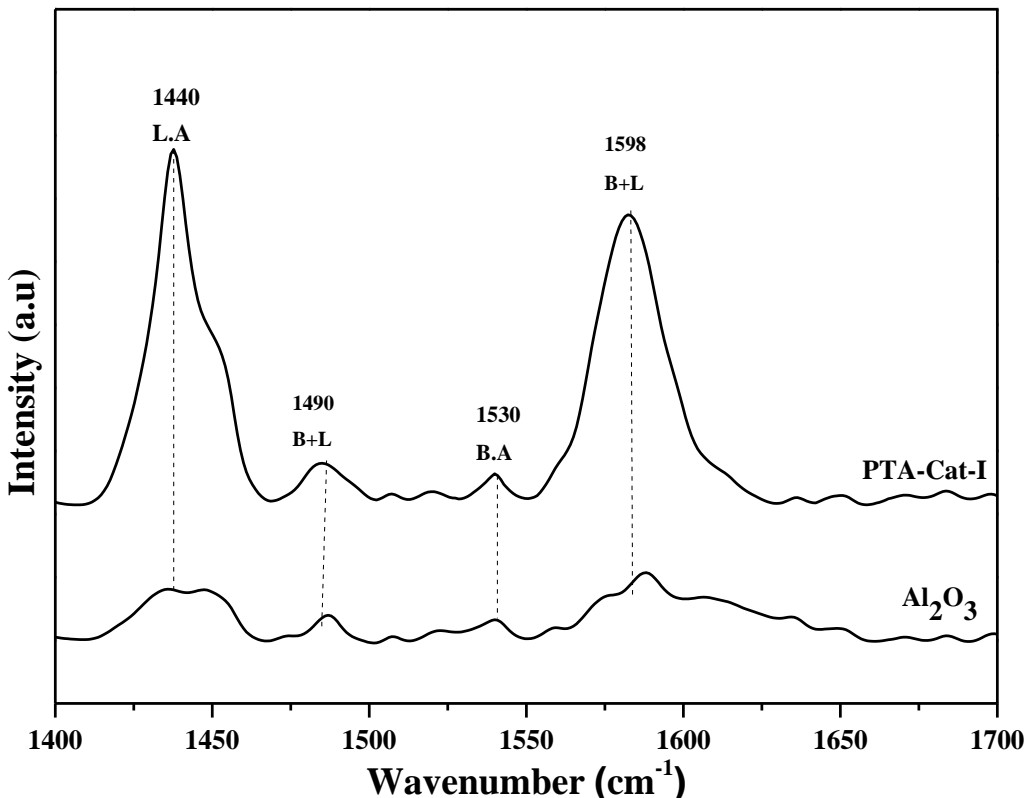

**Figure 4.** Pyridine adsorbed FTIR spectra of $Al_2O_3$ and PTA-Cat-I catalyst samples.

In this research work, we have mainly focused on liquid-phase citral hydrogenation, reaction kinetics, menthol yield/selectivity and other significant operating reaction parameters.

### 3.2. Reactions Study

The catalysts were tested in liquid-phase citronellal isomerization and citral hydrogenation reactions for good catalytic performance and product yield. Furthermore, experimental work was extended and mainly focused on reaction process parameters such as metal type, heteropoly acid impregnation (e.g., acidity or acidic sites) and reaction kinetics for this study. Furthermore, reaction process conditions were optimized in both reactions (e.g., cyclisation citronellal cyclisation and hydrogenation of citral). However, these catalytic reactions were conducted in a stainless steel autoclave reactor under gas pressure ($N_2$ gas for cyclisation and $H_{2\,gas}$ for hydrogenation). Furthermore, activated catalyst and required chemical reagents were added to the reactor and operated for specific periods. To determine the effects of acidity on catalytic activity (e.g., citronellal cyclisation), various heteropoly acid family members (e.g., PTA, SMA, STA and PMA) have been impregnated (20 wt.% loading) on alumina support (e.g., acidic catalysts). Furthermore, various metals (5 wt.%) (e.g., Pd, Pt, Ni, Sn and Cs) were doped over alumina and applied in liquid-phase citral hydrogenation. Additionally, Pd precursors were doped on HPA-alumina supports and applied in liquid-phase citral hydrogenation reaction to obtain a higher yield of menthol.

### 3.2.1. Citronellal Cyclisation to Isopulegol

To obtain higher selectivity and catalytic activity, a series of HPA-supported alumina acidic catalysts (e.g., PTA-$Al_2O_3$, STA-$Al_2O_3$, PMA-$Al_2O_3$ and SMA-$Al_2O_3$) were prepared and examined in a citronellal cyclisation reaction (one step reaction); their reaction performance is presented in Table 2 and Figure 5.

**Table 2.** Reaction performance of Pd-acidic (γ-alumina-supported phosphotungstic acid) catalysts in liquid-phase citronellal cyclisation.

| Catalysts | Conversion [a] | Yield | Selectivity | Ds (±) Isopulegol | Others | K [b] |
|---|---|---|---|---|---|---|
| | (%) | (%) | (%) | (%) | (%) | $min^{-1}$ |
| $Al_2O_3$ | 12 | 2 | 16 | 2 | 10 | 0.003 |
| PTA-Cat-I | 95 | 92 | 97 | 87 | 3 | 0.044 |
| PMA-Cat-I | 89 | 83 | 93 | 84 | 6 | 0.037 |
| STA-Cat-I | 78 | 66 | 85 | 77 | 12 | 0.029 |
| SMA-Cat-I | 62 | 40 | 64 | 58 | 22 | 0.022 |

[a] Reaction conditions: 4.5 mmol citronellal, 0.1 g freshly prepared catalyst, 5 mL $C_7H_8$, T 70 °C, reaction period: 1 h. [b] reaction rate constant.

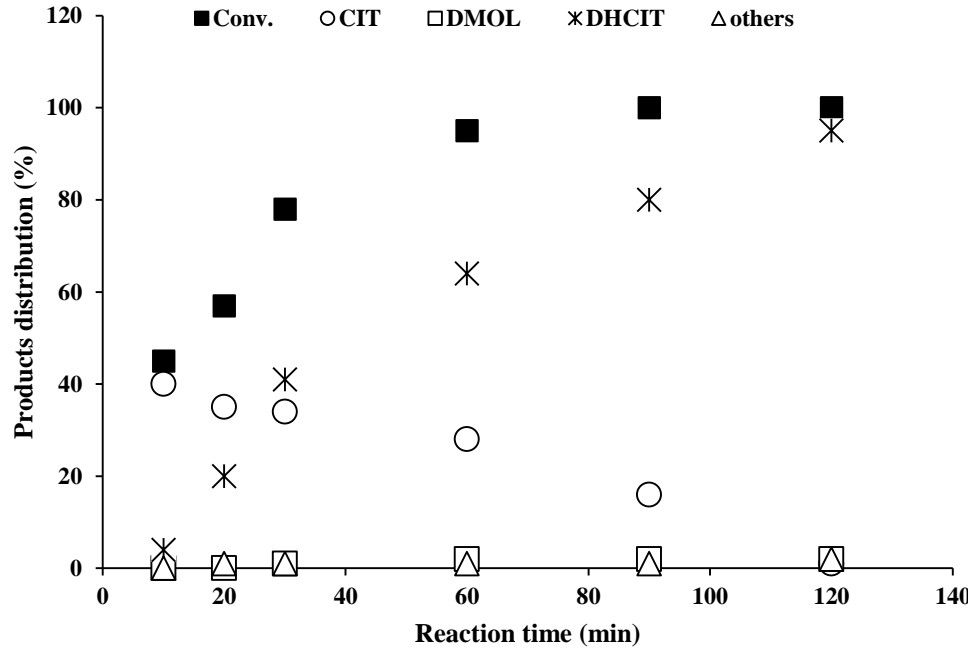

**Figure 5.** Reaction kinetics study and products distribution (%) of 5 wt.(%) Pd-$Al_2O_3$ (Cat-I) catalyst in liquid-phase citral hydrogenation (operating reaction conditions: T = 70 °C, P = 1.0 bar $H_2$ gas).

In the initial stage, ethanol-washed alumina support was tested in cyclisation reaction. Within 1 h of reaction time, alumina transformed citronellal reactant molecules (approximately 12%, lowest reaction rate 0.003 $min^{-1}$) into unwanted side products. The isopulegol formation over this catalyst was negligible. It was observed that the catalytic performance of alumina was significantly improved (e.g., higher reaction rate) with (20 wt.%) impregnation of various heteropoly acids (e.g., PTA, SMA, STA and PMA). However, PTA-$Al_2O_3$ acidic catalyst exhibited the highest catalytic activity (conv. 95% and K = 0.044 $min^{-1}$) and isopulegol selectivity (97%). Furthermore, PMA-$Al_2O_3$ catalyst yielded 92% isopulegol at 89% citronellal conversion (reaction rate k = 0.037 $min^{-1}$) within one hour of reaction time. Additionally, STA-$Al_2O_3$ (e.g., conv. 78%, selectivity 85% and reaction rate 0.029 $min^{-1}$) and SMA-$Al_2O_3$ (e.g., conv. 62%, selectivity 64% and reaction rate 0.022 $min^{-1}$) catalysts were found to be less active and less selective than the PTA-$Al_2O_3$ and STA-$Al_2O_3$ catalyst samples. The reaction kinetics of all prepared catalysts is shown in Figure 5.

The major differences in catalytic activity and selectivity were mainly related with acidity enhancement (creation of Lewis and Bronsted sites) [43], as is clearly shown in Figure 4. The catalytic performances of all catalysts was mainly due to the presence of strong acid sites over the catalyst surface [7,45,46]. The differences in catalytic activity with respect to prepared acidic catalysts were totally dependent on acidity strength. Among

these acidic catalysts, PTA-Al$_2$O$_3$ acidic catalyst was found to be the most active and highly selective acidic catalyst for citronellal cyclisation.

During reaction performance investigation, a few byproducts such as p-menth-3-ene and trans-p-menthane were possibly be formed through dehydration or cracking of isopulegol over the strong acidic sites of the catalyst (173.3). Alumina material possesses a dehydration characteristic, which promotes cracking or dehydration of isopulegol.

### 3.2.2. Citral Hydrogenation to Menthols
### Metal-Supported Alumina Catalysts for Liquid-Phase Citral Hydrogenation

The hydrogenation of citral substrate was conducted in a stainless steel autoclave reactor under optimized conditions. The literature (Scheme 1) and experimental data suggest that synthesis of menthol through citral hydrogenation comprises three consecutive reaction steps: (a) hydrogenation of citral to citronellal (CIT); (b) cyclisation of citronellal (CIT) to isopulegol (IPOL); (c) hydrogenation of isopulegol (IPOL) to menthols (MOL). In the starting research phase, metal (e.g., Pd, Pt, Ni, Cs and Sn)-supported alumina catalysts (e.g., Cat-I (Pd-Al$_2$O$_3$), Cat-II (Pt-Al$_2$O$_3$), Cat-III (Ni-Al$_2$O$_3$), Cat-IV (Cs-Al$_2$O$_3$) and Cat-V (Sn-Al$_2$O$_3$) were applied in the liquid-phase citral hydrogenation reaction under optimized conditions (T = 70 °C and P = 1.0 MPa) (Table 3).

**Table 3.** Reaction performance of bifunctional catalysts in liquid-phase citral hydrogenation.

| Catalysts | Conversion | \multicolumn{7}{c}{Product Distribution Yield (%) [a]} |
|---|---|---|---|---|---|---|---|---|
| | | MOL | CIT | IPOL | DHCIT | DMOL | Others | K [b] |
| | (%) | (%) | (%) | (%) | (%) | (%) | (%) | min$^{-1}$ |
| **Cat-I** | 100 | 0 | 1 | 0 | 95 | 2 | 2 | 0.050 |
| **Cat-II** | 98 | 0 | 63 | 0 | 25 | 3 | 7 | 0.042 |
| **Cat-III** | 92 | 0 | 75 | 0 | 11 | 0 | 6 | 0.037 |
| **Cat-IV** | 74 | 8 | 54 | 5 | 1 | 0 | 6 | 0.017 |
| **Cat-V** | 61 | 17 | 28 | 7 | 4 | 2 | 3 | 0.015 |

[a] Reaction conditions: 4.5 mmol citral, 0.2 g catalyst, 5 mL C$_7$H$_8$, T = 70 °C, P = 1.0 MPa H$_2$, reaction period: 3 h.
[b] reaction rate constant.

It was observed from the experimental study that Pd- and Pt-supported catalysts were more active and had higher selectivity for hydrogenation reactions. However, Cat-I hydrogenated citral substrate to dihydrocitronellal (DHCIT) multiple times and yielded 95% within 1 h. Cat-II and Cat-III yielded dihydrocitronellal (DHCIT) at 25% and 11%, respectively. Additionally, Cat-IV and Cat-V catalysts promoted citral hydrogenation to citronellal and produced 8% and 17% menthols, respectively. In reaction rate comparison, Cat-I possessed a higher reaction rate (0.050 min$^{-1}$) than other catalysts (0.042 to 0.015 min$^{-1}$) (Table 3) and produced 95% dihydrocitronellal as a major product. It was found that Cat-I, II and III samples retain only active metal sites over the alumina surface, which strongly promotes hydrogenation reactions. However, Cat-IV and Cat-V may contain just a few acidic sites (L/B) along with metal sites on the alumina surface, which transforms citral substrate to menthol via multistage hydrogenation–cyclisation–hydrogenation reactions. Table 3 and Figure 5 suggest that these catalysts (e.g., Cat-I, II, III, IV and V) could not produce high yield of menthols via the liquid-phase citral hydrogenation reaction. It is anticipated that the main cause of this (e.g., lower menthol yield) might be the presence of imbalanced acidic sites (L/B) over the catalyst surface. Figure 5 data displays the reaction kinetics of all catalysts in liquid-phase citral hydrogenation. It has been observed that 4.5 mmol (100% reactant molecules) citral substrate molecules were transformed into other hydrogenated products within 2 h.

The chemical route of the citral hydrogenation reaction can be changed based on solvent type, solvent polarity and solubility properties. Metal catalysts promote the transformation of citral reactant molecules to geraniol or nerol alcohol with the use of ethanol or methanol solvents. Acidic sites nature and strength may have affect on the catalytic

activity and selectivity characteristics of solid acid catalyst [45,46]. Similarly, the same metal catalysts can transform citral to citronellal with the use of toluene, hexane benzene and 1-3 dioxane solvents [47,48]. It was observed that most of the metal-supported catalysts (e.g., Cat-I, Cat-II, and Cat-III) were non-acidic in nature and did not promote a citronellal cyclisation reaction. Due to the lack of acidic characteristics (e.g., Cat-I, Cat-II and Cat-III), menthol production from citral hydrogenation was found to be negligible, although, the metal-supported catalysts generally favored hydrogenation reactions, e.g., citral hydrogenation to citronellal and citronellal hydrogenation to dihydrocitronellal, instead of citronellal cyclisation to isopulegol. It has been assumed that metal sites attract aldehyde C=C bonds and promote hydrogenation reactions. In this overall reaction study, toluene solvent was used to obtain a higher yield of hydrogenated and cyclized products. In accordance with previous studies [49–52], the alumina support (Figure 4) may hold a weak Lewis acidic property; therefore, the citronellal cyclisation reaction does not proceed in the absence of strong acids.

Furthermore, the combination of metals (Pd, Pt and Ni) with alumina promoted hydrogenation reactions and a small percentage of side products were formed due to products cracking (e.g., dehydration side reactions) [53,54]. It has been shown in the literature and experimental data that alumina possesses a dehydration property and bonds with OH ions to release them in the form of water [55]. This type of secondary unwanted reaction (e.g., dehydration and cracking) significantly reduces product yield. Similarly, Pd precursor was doped over Si and ammonium zeolite supports and was applied in the liquid-phase citral hydrogenation reaction. Silica- and zeolite-supported catalyst produced only 5% and 24% menthol, respectively. From experimental observations, it was found that the Pd-Zeolite catalyst probably possesses Lewis and Bronsted sites that may promote citronellal cyclisation reactions, as compared to Pd-$Al_2O_3$.

### γ-Alumina-Supported Phosphotungstic Acid-Palladium Catalysts for Liquid-Phase Citral Hydrogenation

In the first stage of experimental work, metal-supported alumina catalysts (Cat-I, Cat-II, Cat-III, Cat-IV and Cat-V) were prepared and applied in the liquid-phase citral hydrogenation reaction. From experimental data, most of these catalysts (e.g., Cat-I. Cat-II and Cat-III) did not promote citronellal cyclisation or menthol production. However, it was found that alumina does not possess any acidic characteristics. Therefore, alumina-supported catalysts were not found to be suitable for citronellal cyclisation (e.g., isopulegol and menthol production). Thus, experimental work was extended and new metal–acid-supported alumina catalysts (e.g., PTA-Cat-I, STA- Cat-I, SMA-Cat-I and PMA-Cat-I) were prepared with the impregnation of (5 wt.%) Pd precursor ($PdCl_2$). Additionally, these prepared catalysts were tested in the hydrogenation of citral to find the drastic effects of various heteropoly acids, along with alumina and palladium. Moreover, Cat-I catalyst did not produce menthol or isopulegol in overall reactions, whereas a specific amount of phosphotungstic acid (20 wt.%) was impregnated that changed the overall performance of Cat-I (e.g., promoted citronellal cyclisation instead of citronellal hydrogenation).

PTA-Cat-I catalyst yielded 45% menthol, 4% isopulegol, 25% dihydrocitronellal, 6% 3, 7-dimethyl-1-octanol and 18% dehydrated or cracked side products. The reaction rate of Cat-I decreased with the impregnation of PTA (e.g., k = 0.050 decreased to 0.038 $min^{-1}$). The reaction rate reduction might be due to the blockage of surface area and pore volume resulting in a decrease in mass transfer/molecule diffusion inside pores of alumina. Furthermore, PMA-Cat-I produced 27% menthol, 23% citronellal, 10% isopulegol, 21% dihydrocitronellal and 18% cracked/dehydrated products (Table 4).

Additionally, the reaction rate of this catalyst (PMA-Cat-I) was 0.032 $min^{-1}$. Similarly, the STA-Cat-I and SMA-Cat-I catalysts were not found to be selective for menthol synthesis. The reaction rates of STA-Cat-I and SMA-Cat-I were 0.029 and 0.024 $min^{-1}$, respectively. Figure 6 shows the reaction kinetics of PTA-Cat-I. The citral substrate was completely transformed into

cyclized and hydrogenated products. With respect to time, citronellal substrate was cyclized into isopulegol and then further hydrogenated to menthol (approx. 45%).

**Table 4.** Reaction performance of multifunctional composite materials (γ-alumina-supported phosphotungstic acid-palladium) in liquid-phase citral hydrogenation.

| Catalysts | Conversion | MOL | CIT | IPOL | DHCIT | DMOL | Others | K [b] |
|---|---|---|---|---|---|---|---|---|
| | (%) | (%) | (%) | (%) | (%) | (%) | (%) | min$^{-1}$ |
| **Cat-I** | 100 | 0 | 1 | 0 | 95 | 2 | 2 | 0.050 |
| **PTA-Cat-I** | 100 | 45 | 2 | 4 | 25 | 6 | 18 | 0.038 |
| **PMA-Cat-I** | 100 | 27 | 23 | 10 | 21 | 1 | 18 | 0.032 |
| **STA-Cat-I** | 78 | 13 | 28 | 12 | 16 | 3 | 6 | 0.029 |
| **SMA-Cat-I** | 67 | 4 | 41 | 3 | 10 | 0 | 9 | 0.024 |

(Column group header: **Product Distribution Yield (%)** [a])

[a] Reaction conditions: 4.5 mmol citral, 0.2 g catalyst, 5 mL $C_7H_8$, T 70 °C, P = 0.5 MPa $H_2$, reaction period: 8 h.
[b] reaction rate constant.

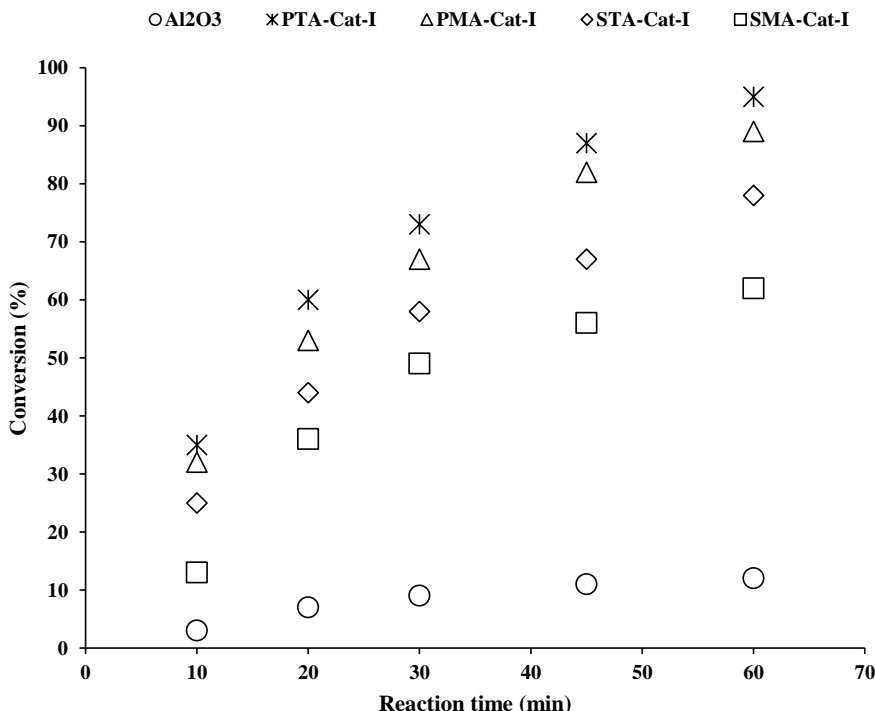

**Figure 6.** Catalytic performance evaluation study of γ-alumina-supported heteropoly acid palladium catalysts (PTA-Cat-I, PMA-Cat-I, STA-Cat-I and STA-Cat-I) in liquid-phase citronellal cyclisation. The reaction was conducted in a glass reactor at 70 °C for 1 h using 4.5 mmol of citronellal substrate and 5 mL toluene.

This optimal menthol yield was achieved after performing several experiments. From experimental data and literature review [25], it was found that phosphotungstic acid is a strong acid and creates strong Lewis and weak Bronsted acid sites over the alumina support during impregnation. According to acidity strength, the order PTA > PMA > STA > SMA was observed with the help of the amine titration technique and reaction performance [25]. Additionally, it was found from several experiments that alumina support was not suitable for menthol synthesis because of its dehydration characteristic. Several experiments were performed in multiple directions to obtain a higher yield of menthol, but all experimental efforts failed because of continuous cracking and dehydration of isopulegol and menthol isomers due to alumina's dehydration characteristic. During the use of Pd precursors such as palladium chloride and palladium acetyl-acetonate in the preparation of metal–acid

catalysts, it was observed that the Pd acetyl-acetonate precursor is a very active for hydrogenation reactions, e.g., hydrogenation of citral to citronellal and hydrogenation of citronellal to dihydrocitronellal. It favors the citronellal hydrogenation reaction instead of citronellal cyclisation. Several experiments were conducted to obtain higher menthol yield (~18.3%), e.g., lowering citronellal hydrogenation and promoting citronellal cyclisation, but we could not achieve this goal. At higher temperatures (>70 °C) and higher pressures (>1.0 to 3.0 MPA), C-C hydrogenation reaction rates were enhanced compared to the cyclisation rate. At lower temperatures (<70 °C) and pressures (<1.0 MPa), the hydrogenation reaction rate decreased down and yielded only 2% menthol. The optimum process parameters (e.g., T = 70 °C, P = 0.5 MPa) were found to obtain the greatest menthol yield (~18.3%), though 54% dihydrocitronellal and 18% cracked side products were also formed, using palladium acetyl acetonate precursor-based catalyst. Similarly, toluene solvent was found to be better than cyclohexane solvent in liquid-phase citral hydrogenation based on catalytic activity and menthol yield (Figure 6).

Furthermore, the research was extended and $PdCl_2$ precursor was doped on acidic supports (e.g., $PTA-Al_2O_3$, $PMA-Al_2O_3$, $SMA-Al_2O_3$ and $STA-Al_2O_3$). This precursor-based catalyst yielded a high amount of menthol (~45%) within 6 to 8 h. $PdCl_2$-based catalysts exhibited slow catalytic activity for citronellal hydrogenation and promoted citronellal cyclisation to isopulegol. The hydrogenation rate of isopulegol to menthol was found to be lower than the citral hydrogenation rate. The optimum conditions of T = 70 °C and P = 0.5 MPa were found in the liquid-phase citral hydrogenation reaction based on menthol production (~45%). Based on menthol yield, palladium chloride precursor was used in the preparation of other bifunctional catalysts (e.g., 20 wt.% of PMA-Cat-I, SMA-Cat-I, STA-Cat-I) using dry impregnation and ion-exchange methods. The ion-exchange method (e.g., using toluene solvent) was found to be better than the dry impregnation method based on catalytic activity and menthol yield. The performance of all catalysts (prepared by ion-exchange method) is shown in Table 4. During experimental observations, higher loading of heteropoly acid (~30 wt.% PTA) over the supported catalyst was not found to be active and selective for menthol production (~19% yield). Generally, it was found that high pressure in the reaction favored C-C hydrogenation, whereas high hydrogen pressure was not suitable for menthol production. However, when the reaction was operated at a high temperature (~100 °C), the C-C chain reaction rate slowed and unwanted cyclic products, such as 4-isopropyl-1-methylcyclohex-ene/ane, were formed. Additionally, when the reaction was conducted in the range of low temperatures (40 to 50 °C), the formation of unwanted cyclic products was reduced (e.g., menthol formation was very low). The formation of these side products (e.g., cracked cyclic compounds) is possibly due to the removal of hydroxyl functional groups from isopulegol and menthol isomers. During the reaction study, it was found that the increase in metal sites (e.g., more Pd loading) may block acidic sites of $PTA-Al_2O_3$ and promote C-C chain reactions (e.g., hydrogenation). More precisely, it was shown that greater loading of Pd metal over acidic support probably blocks its active sites and inhibits cyclisation reaction. Furthermore, this phenomenon promotes the formation of hydrogenated products. However, a higher temperature was not suitable for liquid-phase citral hydrogenation in the presence of acidic catalysts because it decomposed/cracked isopulegol and menthol isomers quickly and decreased menthol yield. Therefore, 20 wt.% loading of phosphotungstic acid was found to be the optimum loading based on catalytic activity, menthol selectivity (~45%), surface area and pore volume. The initial reaction rate (e.g., PTA-Cat-I) for liquid-phase citral hydrogenation (k) was measured (0.038 mmoles.min$^{-1}$) (Table 4 and Figure 7).

Figure 7 reveals the product distribution of the hydrogenation reaction. Based on the catalytic performance, PTA-Cat-I catalyst was observed to be an efficient and moderately selective bifunctional catalyst for menthol synthesis. The formation of other side products (e.g., citronellal ethers, methyl-4-propyl cyclohexane, cyclohexane-1-methyl-4-propylidene and menthatri-ene) was possibly observed due to dehydration, hydrogenolysis and cracking of isopulegol and menthol isomers [52,53]. From experimental data and the reaction

mechanism, it was found that the bifunctional catalyst (PTA-Cat-I) also promoted side reactions (e.g., dehydration and defunctionalization of isopulegol and menthol isomers) [53]. The higher production of menthol (~45%) is probably connected with desired and balanced metal–acid sites. Furthermore, catalytic activity might be directly related to active metal sites and high surface area and pore volume of the catalyst. Furthermore, it was shown that higher heteropoly acid loading (>30 to 60%) would possibly block pores of support and metal sites, and conceivably reduce mass transfer/diffusion of substrate molecules. thereby decreasing catalytic activity and menthol selectivity.

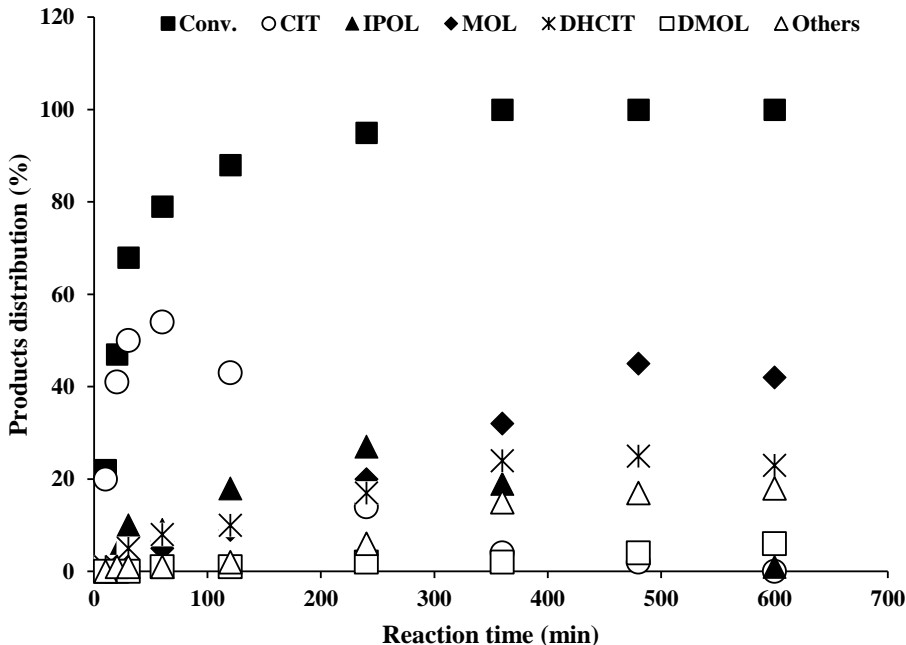

**Figure 7.** Reaction kinetics and product distribution (%) of 5% Pd-20% PTA- Al$_2$O$_3$ (PTA-Cat-I) catalyst in liquid-phase citral hydrogenation (reaction conditions: temperature = 70 °C; pressure = 0.5 MPa H$_2$).

Maki Arvela et al. (2002) prepared Ru-, Rh- and Ni-supported catalysts in citral hydrogenation and produced 92% citronellol [54]; they also studied side product formation and its control. Samrat Mukherjee et al. (2006) determined the solvent's effects on catalytic activity and selectivity parameters in citral hydrogenation and evaluated its mechanisms [46].

Shu-Jen Chiang et al. (2007) prepared NiB/SiO$_2$ catalyst and applied it in citral hydrogenation, producing 98% citronellal/citronellal at low temperature. This is lower than our reported result (100% citronellal, 95% dihydrocitronellal) within one hour [55].

The main reason for higher menthol production might be strong Lewis acid sites (as well as weak Bronsted sites) with desired Pd metal sites. It is well known that citronellal cyclisation is possible in the presence of strong Lewis and weak Bronsted acid sites. There must be a balanced ratio of metal and acid sites (B/L) to obtain a selective and controlled reaction path (e.g., from citral to menthol). Comparing the performance of various Pd-based precursors, PdCl$_2$ precursor (5 wt.%) was found to be more selective for menthol production. The direct conversion of citronellal to menthol seems possible over balanced metal and acid sites (strong Lewis and weak Bronsted acid sites), this would favor the multistage complex reaction mechanism [29,52].

## 4. Conclusions

In this study, alumina (metal)-supported catalysts (e.g., Cat-I (Pd-Al$_2$O$_3$), Cat-II (Pt-Al$_2$O$_3$), Cat-III (Ni-Al$_2$O$_3$), Cat-IV (Sn-Al$_2$O$_3$) and Cat-V (Cs-Al$_2$O$_3$)) and metal–acid-supported alumina (e.g., Cat-I= Pd-Al$_2$O$_3$) (PTA-Cat-I; -PMA-Cat-I; STA-Cat-I, and SMA-Cat-I) were prepared through wet impregnation and ion exchange techniques, respectively,

and applied in citral hydrogenation. The selective bifunctional catalysts were analyzed by PXRD, BET, FTIR, pyridine adsorption and amine titration techniques. The effects of operating parameters were studied and compared with catalytic activity and menthol selectivity. Menthol production was obtained from liquid-phase citral hydrogenation at 0%, 0%, 0%, 8% and 17% using Cat-I, Cat-II, Cat-III, Cat-IV and Cat-V, respectively. Pd-Al$_2$O$_3$ could produce 95% dihydrocitronellal product within 60 min. Furthermore, Pd-doped alumina (Cat-I) catalyst was modified with impregnation of heteropoly acids and obtained menthol production from liquid-phase citral hydrogenation of 45%, 27%, 13% and 4% using PTA-Cat-I, PMA-Cat-I, STA-Cat-I and SMA-Cat-I, respectively, under controlled process conditions (T = 70 °C and 0.5 MPa). High temperatures and pressures possibly promoted hydrogenation reactions and yielded unwanted cyclic products. High rates of unwanted cyclic products were formed due to the dehydration characteristic of the alumina support. A balanced ratio of metal and acid sites catalyzed the conversion of citral to menthol and enhanced its selectivity under optimized process parameters and with selective Pd precursor. High loading of heteropoly acid (>20 wt.%) may block pores and metal sites, causing a decrease in catalytic activity and selectivity. Optimum menthol production may result from the presence of strong Lewis and weak Bronsted acid sites imbalanced with Pd precursor (PdCl$_2$) metal sites. Palladium acetyl acetonate precursor was found to be highly active in hydrogenation reactions, as compared to palladium chloride, but it was not helpful in producing a high amount of menthol. Optimized process conditions (5 wt.% Pd, 20% PTA, T = 70 °C and P = 0.5 MPa) and the ion-exchange technique were more effective in attaining superior menthol selectivity.

**Author Contributions:** A.K.S. (conceptualization, experiments, analysis, and result interpretation, writing manuscript); S.N.-u.S.B. (analysis, result interpretation, and writing manuscript); A.A.S. (result interpretation, manuscript writing, review, and proof reading); A.S.J. (result interpretation, manuscript writing, review); Y.H.P. (conceptualization, design of experiments, provision of laboratory resources, and review); M.-S.C. (conceptualization, design of experiments, provision of laboratory resources, and review); M.A.U., Z.H., G.T.S., A.I., T.H.S. and A.R. (Design of experiments, manuscript writing and English editing services). All authors have read and agreed to the published version of the manuscript.

**Funding:** This research received no external funding.

**Data Availability Statement:** This project was completed at Hanyang University, South Korea during PhD studies under the Higher Education Commission Pakistan HRDI PhD scholarship program.

**Acknowledgments:** We strongly acknowledge the Higher Education Commission Pakistan and Fine Chemical Process Lab, Department of Fusion Chemical Engineering, Hanyang University, Ansan, South Korea for providing PhD scholarship, laboratory facilities and funding for PhD research project.

**Conflicts of Interest:** The authors declare no conflict of interest. The funders had no role in design of the study; in the collection, analyses, or interpretation of data; in the writing of the manuscript; or in the decision to publish the results.

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
