# Peer review of "Design of γ-Alumina-Supported Phosphotungstic Acid-Palladium Bifunctional Catalyst for Catalytic Liquid-Phase Citral Hydrogenation"

_catalysts, doi:10.3390/catal12091069_

Round 1

Reviewer 1 Report

Title of manuscript “Design of γ-alumina supported phosphotungstic acid-palladium bifunctional catalyst for catalytic liquid phase citral hydrogenation is a good draft and technically sounds good.

Recommendation: Publish after minor revisions noted.

Page No: 2, Line 1, Scheme 1, it should be draw the statures. it’s not clear visible.

Page No: 3, Line 84, also describe the HPA.

Page No: 5, Line 204,206,210,219 and 221 check spelling of “Figure”

Line: 136, Impressed , alumina supported metal catalysts/  heteropoly acids (HPA) supported alu- 142 mina based bifunctional metal acids catalysts were prepared freshly ?

Line: 229, Check Figure number is not in Bold.

Line: 332, Check Table 3 format.

A through grammatical check space, commas, spelling mistakes must be performed for the entire manuscript.

Author Response

File Attached. 

Reviewer 2 Report

In the related manuscript (catalysts-1861074), authors have synthesized various bifunctional catalysts involving alumina, a metal and a heteropoly acid (HPA) and tested them for the menthol production. The catalysts were characterized adequately by various techniques. The results arepresented wel using Figures and theory. According to my review, I suggest that manuscript for publication in Catalysts after addressing following issues.

1.       The paper deals with various metal including Pd and other heteropoly acids. Why the title “Design of γ-alumina supported phosphotungstic acid-palladium bifunctional catalyst for catalytic liquid phase citral hy- 3 drogenation” has only Pd and PTA?

2.       Please correct this sentence on page 1 line 42-43:  Citral (3,7 dimethyl-2,6 octadienal) is generally known as unsaturated aldehyde 42 which is presented in the lemongrass oil.

3.       Please check this sentence: Additionally, the current menthol production is obtained via natural and synthetic routes

4.       Line 59-60 : In accordance to previous literature, one pot menthol synthetic reaction route via citral hydrogenation is seemed complex because of improper design (less selective) of catalysts, which results formation of side reactions and 61 decrease menthol production

5.       Please check this sentence: The number of researchers have been designed bifunctional

6.       Line 72: The lower menthol selectivity and regeneration probably be the key parameters of selective bifunctional catalyst design

7.       Please provide full form of PTA

8.       Line 114 : Furthermore, catalyst designing and operating parameters have been studied and compared with catalytic activity and menthol production

9.       Line 136 : In first stage, alumina supported metal catalysts such as 5 wt. % 136 Pd-Al2O3 (Cat-I), 5 wt. % Pt-Al2O3 (Cat-II), 5 wt. % Ni-Al2O3 (Cat-III), 5 wt. % Cs-Al2O3 137 (Cat-IV) and 5 wt. % Sn-Al2O3 (Cat-V). Sentence is incomplete

10.   Please provide the details of preparation. How much precursors were taken? What does the author means by “higher temperatures”?

11.   Line 155: Although, bulk elemental composition was ascertained with the help of inductively coupled plasma-optical emission spectrometry (ICP-OES)

12.   The adsorption-desorption intensity of alumina decreased after impregnation 203 of PTA and Pd metal (Figureure.1). Please check the spelling of Figure

13.   BET adsorption is shown in Figure 2 but discussed before Figure 1. Please address this

14.   There is no discussion on the actual composition of the catalyst. They are determined?

Author Response

File Attached. 

Reviewer 3 Report

This article is devoted to the development of catalysts for the hydrogenation of citral. The article is written in an understandable language, a large amount of experimental data allows us to draw significant conclusions about the work done. The introduction contains a sufficient amount of information and references to understand the issues of the study. The experimental part is well structured, which also helps to understand the logic of the authors. However, there are some points that it is desirable to finalize:

Technical points:

1. Scheme 1. strongly stretched to the sides. The text is visually stretched. It is desirable to make this clearer.

2. "Al2O3" in all cases is written with subscripts. I assume that this could be done by machine translation of the text into English.

3. Unify all drawings.

4. Change "Figureure" to "Figure".

5. "Funding" and "Acknowledgments" require additions.

6. Numbering has shifted in the "References" section. Please check.

Content revision:

1. The use of platinum, palladium and nickel is well known in the hydrogenation of various substances. However, it is not entirely clear why the authors took Cs and Sn.

2. In what oxidation states are the metals in the studied catalysts? This can also affect selectivity and reaction pathways.

3. When describing all figures and tables, it is desirable to refer to literary sources. The authors have a good description, but there is not enough comparison with the literature data.

4. It is desirable to make the conclusions more concise.

5. Please cite other aluminum oxide applications: 10.1007/s13399-022-02587-x.

Author Response

File attached. 

Round 2

Reviewer 2 Report

The authors have addressed all my concerns. The paper can be accepted.